# Exogenous capture of visual spatial attention by olfactory-trigeminal stimuli

**Matthieu Ischer[1,2], Géraldine Coppin[1,2,3], Axel De Marles[1], Myriam Essellier[1,2], Christelle Porcherot[4], Isabelle Cayeux[4], Christian Margot[4], David Sander[1,2], Sylvain Delplanque[1,2]***

**1** Swiss Center for Affective Sciences, University of Geneva, Geneva, Switzerland, **2** Department of Psychology, University of Geneva, Geneva, Switzerland, **3** Swiss Distance University Institute (UniDistance/FernUni), Brig, Switzerland, **4** Firmenich, S.A., Geneva, Switzerland

* sylvain.delplanque@unige.ch

**Data Availability Statement:** All the data upon which the statistical analyses were performed are available at: https://osf.io/6vpzh/?view_only= ef734433920c41bb84da239ed798f8db#show_login.

## Abstract

The extent to which a nasal whiff of scent can exogenously orient visual spatial attention remains poorly understood in humans. In a series of seven studies, we investigated the existence of an exogenous capture of visual spatial attention by purely trigeminal (i.e., $CO_2$) and both olfactory and trigeminal stimuli (i.e., eucalyptol). We chose these stimuli because they activate the trigeminal system which can be considered as an alert system and are thus supposedly relevant for the individual, and thus prone to capture attention. We used them as lateralized cues in a variant of a visual spatial cueing paradigm. In valid trials, trigeminal cues and visual targets were presented on the same side whereas in invalid trials they were presented on opposite sides. To characterize the dynamics of the cross-modal attentional capture, we manipulated the interval between the onset of the trigeminal cues and the visual targets (from 580 to 1870 ms). Reaction times in trigeminal valid trials were shorter than all other trials, but only when this interval was around 680 or 1170 ms for $CO_2$ and around 610 ms for eucalyptol. This result reflects that both pure trigeminal and olfactory-trigeminal stimuli can exogenously capture humans' spatial visual attention. We discuss the importance of considering the dynamics of this cross-modal attentional capture.

## Introduction

Many species display odor-guided spatial navigation. This has been demonstrated in insects [1], fishes [2], birds [3] but also in mammals (e.g., mice [4]; rats [5]; or dogs [6]). This ability is crucial considering that olfaction is essential for detecting and identifying food items, suitable partners and potentially harmful volatile substances [7, 8].

Evidence showing that humans are able to voluntarily (endogenously) pay attention to the spatial location of odors [9] is rare. Humans can track an outdoor whiff of scent, improve this ability with training and display behaviors similar to dogs when the scent changes direction [10]. Moreover, after being disoriented, they can find their way back to a location based only on chemical stimuli [11]. But can this spatial information be exogenously extracted and used

**Funding:** This research was supported by the National Center of Competence in Research (NCCR) for the Affective Sciences, financed by a grant from the Swiss National Science Foundation hosted by the University of Geneva, and was also supported by a research grant from Firmenich, SA, to David Sander and Patrik Vuilleumier. The funder provided support in the form of salaries for author MI and provided material to conduct the experiments but did not have any additional role in the study design, data collection and analysis, decision to publish, or preparation of the manuscript.

**Competing interests:** The authors declared no potential conflicts of interests with respect to the authorship and/or publication of this article. The commercial affiliation of CP, IC & CM does not alter our adherence to PLOS ONE policies on sharing data and materials.

to orient attention in other sensory modalities, e.g. vision? We hypothesized it can, based on three main pieces of evidence.

First, we perceive chemical stimuli in the environment [12, 13] through the olfactory and the trigeminal systems [14]. The trigeminal component of a chemical stimulus enables the differentiation of a stimulation coming from the left vs. the right nostril [12, 15–17]. Although still unclear, inter-nostril comparisons [18] are precisely the process purportedly underlying voluntary scent-tracking [10]. Consequently, the olfactory trigeminal system allows a spatial lateralized information, which could be used to orientate attention. Second, the trigeminal system conveys sensations of pain, cooling, warmth and intensity [19, 20]. In this regard, this system can be considered as an alert one, protecting the organism against potentially dangerous chemical stimuli in the environment [21]. Such stimuli are considered salient ones, which are precisely known to capture attention [22–24]. Third, there is a common pool of attentional resources for the processing of stimuli from all sensory modalities and chemical senses are no exception [25, 26]. Noteworthy, the presence of an ambient odor of lemon (which activates both the olfactory and trigeminal systems) in the environment will facilitate visual search of a lemon picture [27].

The above-mentioned results suggest that there could be an exogenous capture of visual attention by pure nasal trigeminal or by olfactory-trigeminal stimulations. But the questions we asked are more specific: Will attention be exogenously captured by an olfactory-trigeminal stimulus delivered at a given spatial location? If so, will the processing of a visual stimulus (a target) presented at the same location be impacted? To investigate these questions, it is essential to use an appropriate methodology (see [28] for a discussion on this aspect). One of the most used paradigms to test the exogenous capture of attention within and between modalities is the spatial cueing task [29]. In the visual version of the paradigm, a cue and a target are presented successively to the participant on one side or the other of the visual field. The participant's task is to detect a predefined feature of the target (e.g., its horizontal vs. vertical orientation) as fast and accurately as possible independently of its location. Trials in which cues and targets are on the same side are considered valid. Conversely, invalid trials are those in which cues and targets are on opposite sides. An attentional capture by the cue is demonstrated if reaction times (RTs) in valid trials are shorter than in invalid and control (i.e., when no cue is presented) trials. This result was found in a robust way within the visual, auditory and tactile modalities but also in a less documented way between these modalities [26, 30–35].

A first piece of evidence suggests that this result could also be found when using olfactory-trigeminal stimuli as cues and visual stimuli as targets. Using these cues and targets, Wudarczyk and colleagues [36] showed that, although participants were not statistically faster for valid trials than for invalid ones, they were more accurate in the valid condition than in the invalid one. These results consequently suggest that lateralized trigeminal cues can influence visual spatial attention. We believe that the demonstration of an exogenous capture of visual attention by trigeminal stimulation on reaction times could be provided by changing one critical aspect of the experimental design: the Stimulus Onset Asynchrony (SOA).

Our reasoning is the following: attentional capture by stimuli is observed in very specific time windows [37]. In classical spatial cueing tasks, this interval corresponds to the time between the cue onset and the target onset–i.e., the SOA. This SOA is known to be critical to observe attentional capture effects both within [38] and between [39] sensory modalities [32, 33]. In Wudarczyck et al.'s experiment [36], SOA was 500 ms, which might not be optimal to observe an attentional capture with trigeminal (e.g., $CO_2$) cues. Effective SOAs depend mainly on the sensory modality. Yet, the chemosensory related sensory processes are known to be slow, which is reflected by long reaction times during detection and categorization tasks [40–44], and the attentional capture effect may therefore require longer SOAs.

This research therefore aims at i) determining whether a lateralized olfactory-trigeminal stimulus exogenously orientates visual spatial attention and ii) providing a first indication on the time window in which this attentional capture occurs. To do so, the trigeminal system was stimulated with $CO_2$ and eucalyptol. $CO_2$ gas provokes stinging and/or pungency sensations in the nose [45]. Since the $CO_2$ almost exclusively stimulates the trigeminal system [20, 46], it provides a rare and ideal candidate to investigate trigeminal interactions with visual spatial attention. Thus, in a first step, we used a purely trigeminal stimulation, which prevented us from introducing potential interactions between the trigeminal and the olfactory systems. This allowed us to put ourselves in the "ideal" experimental situation, i.e. restricting the observed effects exclusively to an activation of the trigeminal system. We also used eucalyptol, which is more likely to be found in natural situations. Eucalyptol activates both the olfactory and trigeminal system [20] and is widely used in clinical research to assess trigeminal sensitivity [15]. Thus, in a second step, we considered the "ecological" argument, i.e. that $CO_2$ at the used concentrations is not a very common stimulation in nature. Consequently, the phenomenon of attentional capture that we expected to observe with $CO_2$ could be artificial, which would have reduced the scope of the results. We therefore decided to use a mixed olfactory and trigeminal compound that is easily found in our daily lives in order to extend the conclusions brought by $CO_2$. The trigeminal system is activated via several types of receptors (e.g.,20] that are more or less specific to certain compounds. It seemed interesting to study attentional capture effects for at least two types of receptors/molecules to rule out the possibility that these attentional effects are solely and quite artificially produced by $CO_2$. We hypothesized that as soon as a compound or gas has the property to stimulate the trigeminal system, we should observe an attentional capture, i.e., that reaction times of valid trial cued by pure trigeminal ($CO_2$) and olfactory-trigeminal (eucalyptol) stimulations will be shorter than the reaction times for all other trials.

In seven studies, we used either $CO_2$ or eucalyptol as cues and visual stimuli as targets in a cross-modal spatial cueing task. Through our studies, we used SOAs ranging from approximatively 580 ms to 1870 ms. To facilitate the presentation of the studies, we named experiments according to i) a letter representing the type of trigeminal stimulation (C for $CO_2$ and E for eucalyptol) and ii) the SOA used. We hypothesized that as soon as a compound or gas has the property to stimulate the trigeminal system, we should observe an attentional capture, i.e., that reaction times of valid trial cued by pure trigeminal ($CO_2$) and olfactory-trigeminal (eucalyptol) stimulations will be shorter than the reaction times for all other trials.

## Materials and methods

All participants were recruited among undergraduate and graduate students of the University of Geneva and their acquaintances. Characteristics of the sample size, gender and age of the samples are presented in Table 1. A priori determination of sample sizes was not possible since, to our knowledge, no study has reported different reaction times of valid trials cued by trigeminal stimulations being different from reaction times for all other trials. Rather, we calculated the effect sizes ($\eta^2$) required in the main contrast analysis for each experiment using G*Power 3 [47], given $\alpha = .05$ and power $(1 - \beta) = .8$. Required $\eta^2$s: C580 = .24, C680 = .25, C1170 = .25, C1870 = .31, E610 = .25, E830 = .25 and E1120 = .31.

All participants had self-reported normal or corrected to normal vision, self-reported normal olfaction, and no reported psychiatric nor neurological diseases. They received money or partial course credits for their participation in the studies, which were approved by the Faculty of Psychology and Educational Sciences Ethics committee of the University of Geneva. A signed informed consent was obtained from all participants upon their arrival in the laboratory.

**Table 1. Name of the experiment, trigeminal compound, SOA used, sample size (number of female) mean age (± SD) for all studies.**

| Name | Compound | SOA (ms) | Sample size | Age |
|------|----------|----------|-------------|-----|
| C580 | CO2 | 580 | 28 (11) | 26.0 ± 4.4 |
| C680 | CO2 | 680 | 27 (19) | 28.4 ± 6.1 |
| C1170 | CO2 | 1170 | 27 (15) | 27 ± 4.9 |
| C1870 | CO2 | 1870 | 20 (13) | 26.8 ± 4.5 |
| E610 | Eucalyptol | 610 | 27 (18) | 23.0 ± 3.6 |
| E830 | Eucalyptol | 830 | 27 (23) | 23.1 ± 4.3 |
| E1120 | Eucalyptol | 1120 | 25 (19) | 22.7 ± 4.2 |

### Olfactory-trigeminal stimuli

The olfactory-trigeminal stimuli (that we will call trigeminal for simplicity) consisted of $CO_2$ or eucalyptol, depending on the study. The $CO_2$ gas was diluted in cleaned air to obtain an individual concentration determined according to the participant's perception and lateralization performance evaluated in two tasks (described below). The eucalyptol stimuli were made of 7 mL of pure eucalyptol injected into tampons of cylindric felt-tip pens (14 cm long, inner diameter 1.3 cm) positioned in glass vials. Control trials were composed of cleaned air.

All control and trigeminal stimuli were delivered via a custom-made olfactometer which is fully described in Ischer et al. [48],. This device reliably delivers various kinds of chemical stimulations over multiple trials, without contamination from one trial to another, at known latencies, and without additional noise or tactile stimulation [23, 48–50]. $CO_2$ and eucalyptol were embedded in respectively 1 $L.min^{-1}$ and in 0.7 $L.min^{-1}$ constant and filtered airstream.

Control and trigeminal stimulations were delivered directly into the nasal cavity through sterilized stainless steel tips of 4 mm diameter, which were partially introduced into the nostrils (~1 cm, Fig 1A and 1B). The aim was to directly deliver the stimulations in the direction of the anterior portion of the nasal cavity, which is the most sensitive area to trigeminal stimulations [51]. We use E-prime software to present the visual stimuli, to control the olfactometer and to record participant's behavioral responses (Psychology Software Tools Inc., Pittsburg, PA).

### Physiological recordings

Respiration and electro-oculography activities were recorded (1 KHz sampling rate) using the MP150 Biopac Systems (Santa Barbara, CA) in order to monitor participants' compliance with the breathing and gaze position instructions during the experiment. The respiratory activity was recorded through a 2.5 mm tube (interior diameter) taped on the stainless steel tip used to deliver olfactory stimulations and connected to a differential pressure transducer (Biopac TSD160A) to continuously record variations in nostrils airflow [52]. Horizontal electro-oculographical (EOG) activity was assessed using Beckman Ag-AgCl electrodes (8-mm diameter active area) placed at 2 centimeters posterior to the outer canthus of each eye. A supplementary reference electrode was positioned in the middle of the upperpart of the forehead.

### Procedure

Participants first provided their written consent and filled a form to ensure the respect of inclusion criteria and to collect demographic data. The studies using $CO_2$ consisted in 4 different phases: (1) an individual determination of $CO_2$ concentration; (2) a lateralization task; (3) the calibration of the EOG; and (4) the cross-modal spatial cueing task. Studies using pure eucalyptol were composed of phases 2, 3 and 4 only. The entire experimental session took approximately 70 minutes with the $CO_2$ and 60 minutes with the eucalyptol (Fig 1C).

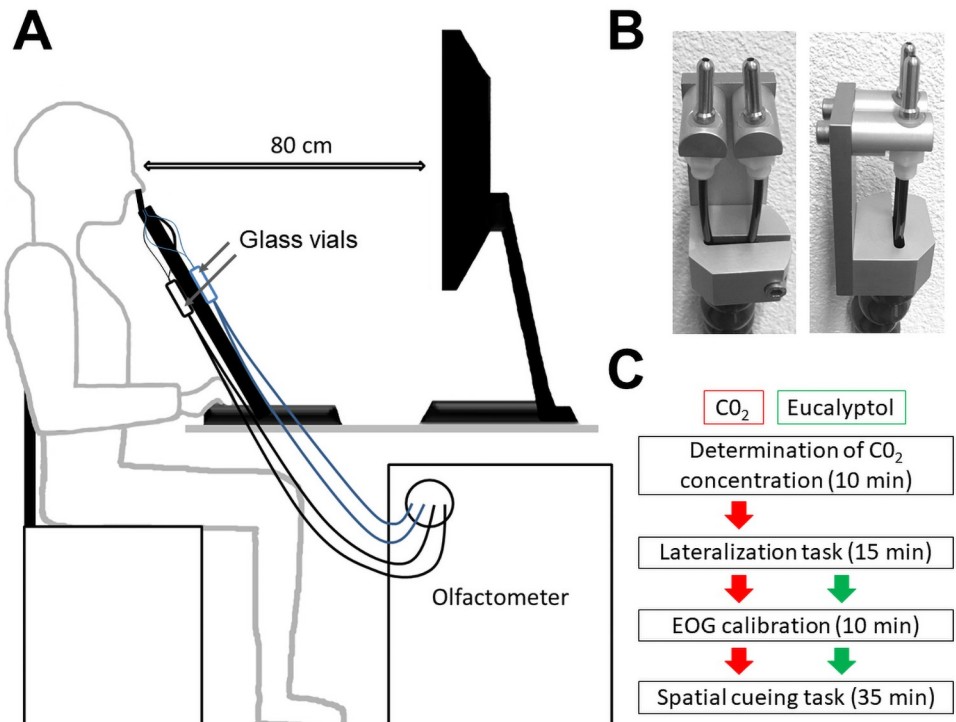

**Fig 1.** A) Drawing of the experimental set up. The color of the tubes differentiates the part that carries air (control condition) from the part that carries trigeminal stimuli. B) Stainless steel tips. Left panel = front view, right panel = lateral view C) Summary of experimental procedure for all studies that used $CO_2$ (in red) and Eucalyptol (in green).

**Determination of CO2 concentration.** This procedure allowed participants to familiarize with the set up and to choose the concentration they found clearly perceptible but not painful. In parallel, participants were instructed and trained on how to proceed with the cued sniffing procedure, which was going to be used during the entire study in order to avoid any interference in the nasal perception due to the natural flow [53]. For each trial (manually triggered), a countdown of 4 seconds was presented on the computer screen, followed by the trigeminal stimulation delivered at zero during 500 ms. Participants were requested to hold their breath from the "1" of the countdown until the end of the trial when they can breathe evenly. The $CO_2$ concentration was set first at the lowest concentration (3% v/v). The mix is first created in a mixing carboy by manually adjusting the input flow of pure CO2 into the cleaned airflow. The $CO_2$ concentration was then increased (or decreased) and adjusted to reach a reported comfortable, not painful but clear perception.

**Lateralization task.** Participants performed a lateralization performance task to assess their ability to lateralize the trigeminal stimulations ($CO_2$ or eucalyptol depending of the study) delivered at the chosen concentration. Twenty trigeminal and 20 control (cleaned air) stimulations embedded in a constant 1 L.min$^{-1}$ airflow were randomly but equally delivered in the right or left nostril for 500 ms. For each trial (every 15 s), participants were requested to indicate if they felt any change in the left or right nostril by pressing the left or right arrow of a keyboard, accordingly. They were requested not to press if they could not feel any change at all. The success criteria was determined using the binomial distribution (one-tailed, $\alpha < .05$, probability of guessing p = .5, n = 10) to ensure that participants lateralized the trigeminal stimulation above chance. Participants obtaining less than 7 correct judgments out of 10

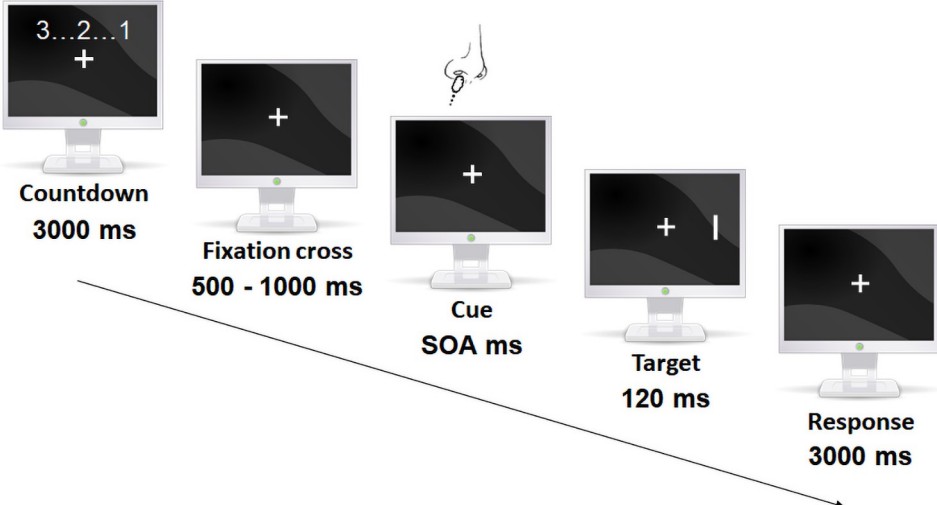

**Fig 2. Cross-modal adaptation of a spatial cueing task [29].** Each trial started with a countdown (3 s) at the end of which participants held their breath. It was followed by the presentation of a fixation cross (500–1000 ms) that participants had to fix. The olfactory cue was then delivered in one nostril during a period which length depended on the SOA. The target appeared for 120 ms on the right or left side of the screen. Participants had to indicate as fast and accurate as possible the orientation of the target (horizontal vs. vertical). Participants breathed evenly after their response. In valid trials, cues and targets appeared on the same side and on opposite sides in invalid trials.

possible correct answers for $CO_2$ trials were not included in the main experiment (n = 13 across all studies).

**Electro-oculographic calibration task.** We recorded eye movements using electro-oculography (EOG) to control for possible horizontal saccades that may occur during the spatial cueing task if the participant does not comply with the instruction to maintain their gaze fixed. To perform this crucial control [30], we first determined what would be the EOG signal corresponding to a saccade performed from the fixation cross to the lateralized targets presented during the spatial cueing task. Participants were requested to look at a white fixation cross presented in the middle of the black screen for 5 seconds and orientate their gaze at the white targets randomly appearing for 1 second on the right (10 trials) or left side (10 trials) at the same location as in the main spatial cueing task. Targets were rectangles of 0.5 cm by 0.1 cm presented either horizontally or vertically. The distance between the fixation cross and the center of each target was 15 cm, resulting in a visual angle of about 10˚ for the participant positioned at 80 cm of the screen.

**Cross-modal spatial cueing task.** In this cross-modal adaptation of a classical spatial cuing task [29], an olfactory cue was delivered either in the right or left nostril prior to a visual target presented during 120 ms either in the right or the left visual field (Fig 2).

In valid trials, cues and targets were delivered on the same side. In invalid trials, they were delivered on opposite sides. SOA values were a priori fixed and increased depending on the study and a jitter appeared due to the loading of the olfactometer's control libraries by the stimulus presentation software (E-prime). This unexpected but measurable jitter was maintained for all studies since it prevented the participant to anticipate the appearance of the target. Resulting mean SOAs and standard deviation for each experiment are presented Table 2. The SOA also determined the duration of the olfactory cue. Nasal cues, equally distributed on both sides, were non-informative of the target location, also equally distributed on both sides (i.e., exogenous cueing). For half of the trial, the cue consists of a trigeminal stimulation. For

**Table 2. Mean (± SD) stimulus onset asynchrony (SOA) for all studies.**

| Study | SOA (ms) | ± SD |
|-------|----------|------|
| C580 | 585 | 53 |
| C680 | 683 | 168 |
| C1170 | 1169 | 117 |
| C1870 | 1870 | 118 |
| E610 | 612 | 37 |
| E830 | 832 | 62 |
| E1120 | 1124 | 63 |

the other half, a valve delivering the same clean air as during inter-stimulus intervals was delivered as a cue. These trials constitute control conditions that mimic every potential other physical changes (e.g., sound, tactile, temperature changes) due to the cueing. Using their right hand, participants were requested to indicate as fast and accurately as possible the orientation of the target using the left arrow key (i.e., horizontal) or the down arrow key (i.e., vertical), both orientations being presented in equal proportions. In total, 64 trials were delivered randomly to each participant (16 for each combination of $CO_2$ or eucalyptol, air, valid and invalid conditions).

## Electrophysiological data analyses

The respiratory flow signal was low-pass filtered at 1 Hz and the flow amplitude calculated as the mean flow values between the cue onset and the target onset. The EOG signal was low-pass filtered at 20 Hz, the activity corresponding to the position of the gaze on the fixation cross (baseline) was calculated as the mean voltage obtained in the calibration phase during the 200 ms preceding the appearance of the target. Typical EOG activities linked to saccades toward left or right targets were determined during the calibration phase by calculating the averaged extremum voltage values calculated while participants were requested to voluntarily look at right (10 trials) and left targets (10 trials). During the cross-modal spatial cueing tasks, gaze position was controlled by averaging extremum voltage values between cue onset and target onset for all relevant trials (air left, air right, trigeminal left, trigeminal right). Those values were then compared with mean EOG activities obtained during both baseline and target fixation periods of the EOG calibration task.

## Statistical analyses

To test our a priori hypothesis, we performed planned contrast comparisons on correct reaction times (RT in ms) to evaluate our specific predictions [54–56] using Statistica (v.13). More specifically, visual attentional capture by trigeminal stimuli would result in shorter reaction times for trigeminal valid trials than for all other trials. Contrast cell weights were as follows: −1 for the invalid air cell, −1 for the valid air cell, −1 for the for the invalid trigeminal cell, and +3 for valid trigeminal cell. We report p values and partial $\eta^2$ as estimates of effect size and their 90% confidence interval (CI). We conducted these very same analyses on response accuracy (ACC) since a better accuracy has been previously reported in valid trigeminal trials as compared to other trials [36]. We also calculated and performed these analyses on the "inverse efficiency score" (RT/ACC, [57]), a measure that combines speed and error to investigate whether the attentional capture resulted in an increase in efficiency of the response (that cannot be derived from reaction time or accuracy alone). Individual respiratory flow values obtained separately for trigeminal, air, left nostril and right nostril stimulations were submitted

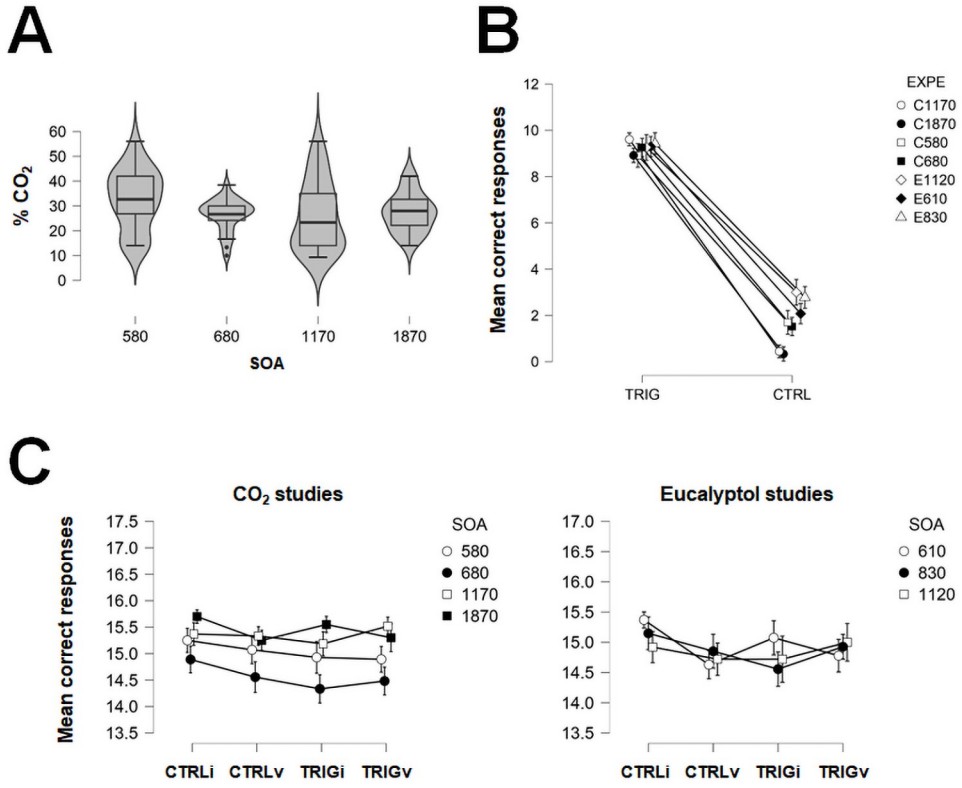

**Fig 3.** A) Boxplots and violin plots of $CO_2$ concentrations (% in air) used for the spatial cueing tasks with CO2 as a function of SOA (in ms). B) Mean correct responses (± SEM) to the lateralization tasks for all types of stimulations, for all studies. TRIG = trigeminal trials, CTRL = control trials (clean air), EXPE = experiment name. C) Mean correct responses (± SEM) to the different conditions of the cross-modal spatial cueing tasks, for CO2 and Eucalyptol studies, separately. TRIG = trigeminal trials, CTRL = control trials, i = invalid trials and v = valid trials.

to t-tests against zero (i.e., no flow). EOG values obtained in baseline, air left, air right, trigeminal left, trigeminal right, left target and right target conditions were submitted to a repeated measures analyses of variance (ANOVAs). Pairwise post-hoc tukey HSD tests were further performed when relevant. An alpha level of.05 was chosen for all the statistical analyses performed.

## Results

### Determination of $CO_2$ concentration

The average percentage of $CO_2$ in the air was 28.28% ± 1.08 (Mean ± SEM). The means' concentration of $CO_2$ are presented Fig 3A as a function of SOA. This concentration was not statistically different between studies, $F(3,98) = 2.35$, $p = .08$, $\eta^2 = .07$, 90% CI = [0,.14].

### Lateralization task

As expected after selecting participants who obtained a minimum of 7 correct lateralization out of 10 possible for each nostril, mean correct lateralization performances for $CO_2$ and eucalyptol stimulations were very high, M = 9.25, SD = 0.88. The number of correct lateralization did not statistically differ across all studies, $F(6,174) = 2.09$, $p = .056$, $\eta^2 = .07$, 90% CI = [0,.10]. By contrast, participants' performance to lateralize control stimulations never reached the statistical criterion of 7 correct responses, whatever the study and the nostril stimulated

(M = 1.78, SD = 1.69). Means of correct trigeminal and control stimulations obtained for all participants are presented Fig 3B as a function of the experiment.

## Cross-modal spatial cueing task

**Response accuracy.** On average, participants were able to indicate the orientation of the target with a very high accuracy (M = 14.99, SD = 0.96, Fig 3C). Response accuracy did not statistically differ between trigeminal valid trials as compared to all other trials, whatever the study, all $F_s$ < 1.53, $p_s$ >.23, $\eta^2_s$ < .05.

**Reaction time.** Table 3 provides participants mean reaction times (± SEM) in every condition for the seven studies.

**Attentional capture by $CO_2$.** Contrasts analyses did not reveal any statistically significant difference between valid trigeminal trials and all other trials for SOAs around 580 ms, $F(1,27)$ = 2.41, $p$ = .13, $\eta^2$ = .08, 90% CI = [0,.26], and 1870 ms, $F(1,19)$ = 0.01, $p$ = .909, $\eta^2$ < .01, 90% CI = [0,.04]. Planned comparisons revealed statistically significant shorter reaction times for valid trigeminal trials than all other types of trials for a SOA of around 680 ms, mean difference = 24 ± 9 ms (SEM), $F(1,26)$ = 7.01, $p$ = .014, $\eta^2$ = .21, 90% CI = [.03,.41], and 1170 ms, (30 ± 6 ms), $F(1,26)$ = 22.03, $p$ < .001, $\eta^2$ = .46, 90% CI = [.21,.61] (Fig 4A).

**Attentional capture by eucalyptol.** Contrasts analyses did not reveal any statistically significant difference between valid trigeminal trials and all other trials with a SOA around 830 ms, $F(1,26)$ = 0.52, $p$ = .48, $\eta^2$ = .02, 90% CI = [0,.17], and 1120 ms, $F(1,24)$ = 0.76, $p$ = .39, $\eta^2$ = .03, 90% CI = [0,.20]. Reaction times to trigeminal valid trials were statistically shorter than for other trials with a SOA around 610 ms, (42 ± 14 ms), $F(1,26)$ = 9.00, $p$ = .006, $\eta^2$ = .26, 90% CI = [.05,.45] Fig 4B.

## Cost effects analysis

To examine possible cost effects that might have been observed for invalid trigeminal trials (attention retained at the invalid location), we performed another planned contrast comparison on correct reaction times (RT in ms). Contrast cell weights were as follows: −1 for the invalid air cell, −1 for the valid air cell, + 2 for the for the invalid trigeminal cell, and 0 for valid trigeminal cell. For the three experiments that revealed an attentional capture, response times did not statistically differ between trigeminal invalid trials as compared to all other control trials, all Fs < 1.84, $p_s$ >.18, $\eta^2_s$ < .07.

**Table 3. Mean (M) reaction times (± SEM) for the trigeminal valid (TRIG valid) and all other conditions together (Others) obtained during the cross-modal spatial cueing tasks for all studies.**

|  | Others | | TRIG valid | |
|---|---|---|---|---|
| **Study** | *M* | *SEM* | *M* | *SEM* |
| **C580** | 735 | 36 | 720 | 35 |
| **C680** | *699* | *22* | *675* | *24* |
| **C1170** | *730* | *28* | *699* | *26* |
| **C1870** | 757 | 31 | 760 | 35 |
| **E610** | *694* | *30* | *652* | *26* |
| **E830** | 678 | 20 | 684 | 22 |
| **E1120** | 680 | 29 | 690 | 35 |

Statistically significant different means are presented in bold italics.

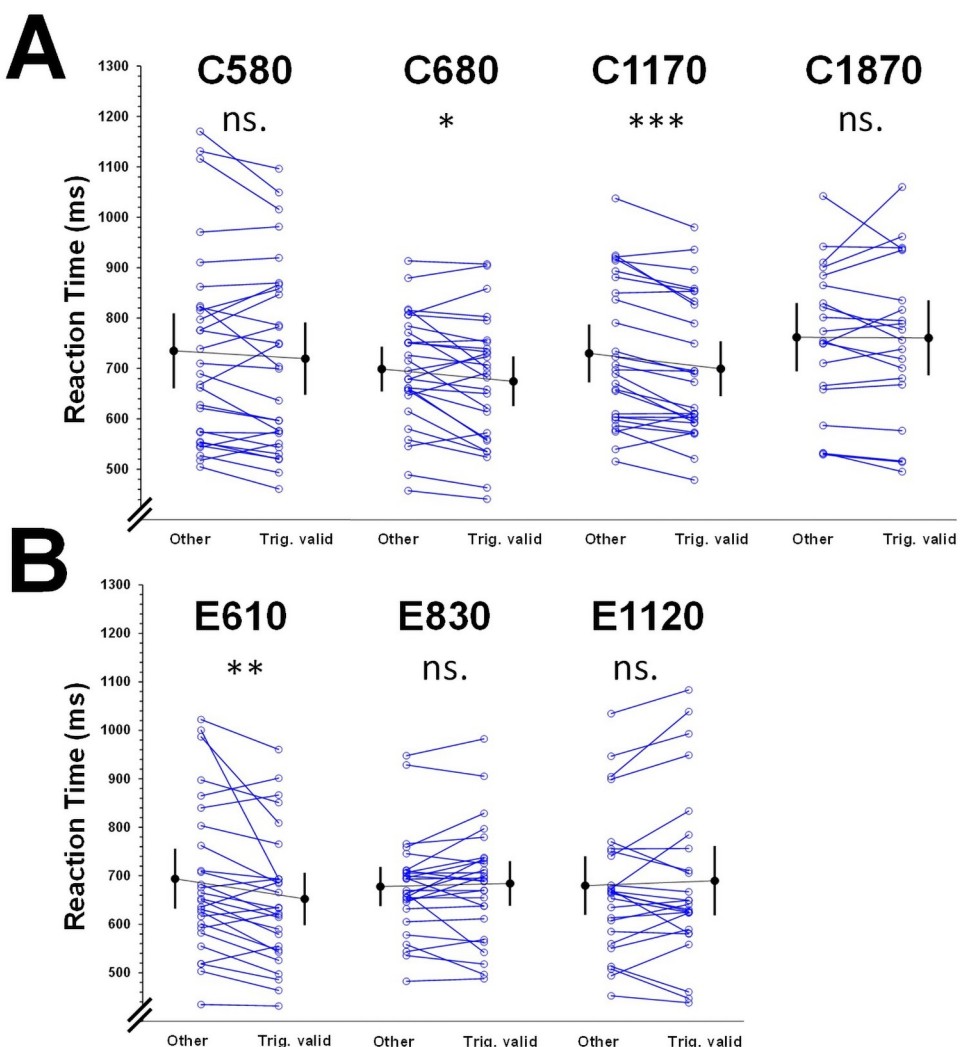

**Fig 4. Mean (black dot) correct reaction times with their 95% CI, as well as individual correct reactions times (blue circles) obtained for the seven studies in response to targets in valid trials after cueing with trigeminal stimulations (trig. valid) and all other trial types averaged (other).** Panel A is for $CO_2$ and panel B for eucalyptol. Mean SOAs are reported on the top of each study's graph. * = p < .05, ** = p < .01, *** = p < .001 and ns. = no statistical significance for the main reported contrasts.

## Efficiency

Inverse efficiency scores were smaller, indicating an increase in efficiency in response to valid trigeminal trials when compared to all other trials in C1170 [F(1,24) = 7.840, p = .009, $\eta^2$ = .24, 90% CI [.03,.44]] and E610 [F(1,26) = 5.88, p = .022, $\eta^2$ = 0.18, 90% CI [.01,.38]] experiments. In the other five experiments, the analyses did not reveal any statistically significant difference (all $F_s$ < 1.67, $p_s$ >.20).

**Experimental controls.** Respiratory activity measured during the cue-target period did not statistically differ from 0 (T-tests against mean, -1.56 < all $t_s$ > 1.87, all $p_s$ >.07), indicating participants held their breath as requested. For all studies, averaged EOG activity statistically differed during baseline, cue-target periods and voluntary saccades (Fig 5), all $F_s$ > 121.88, $p_s$ < .001, $\eta^2_s$ >.84. More precisely, Tukey HSD post hoc comparisons revealed that the average EOG activity during the cue-target period did not statistically differ from the baseline

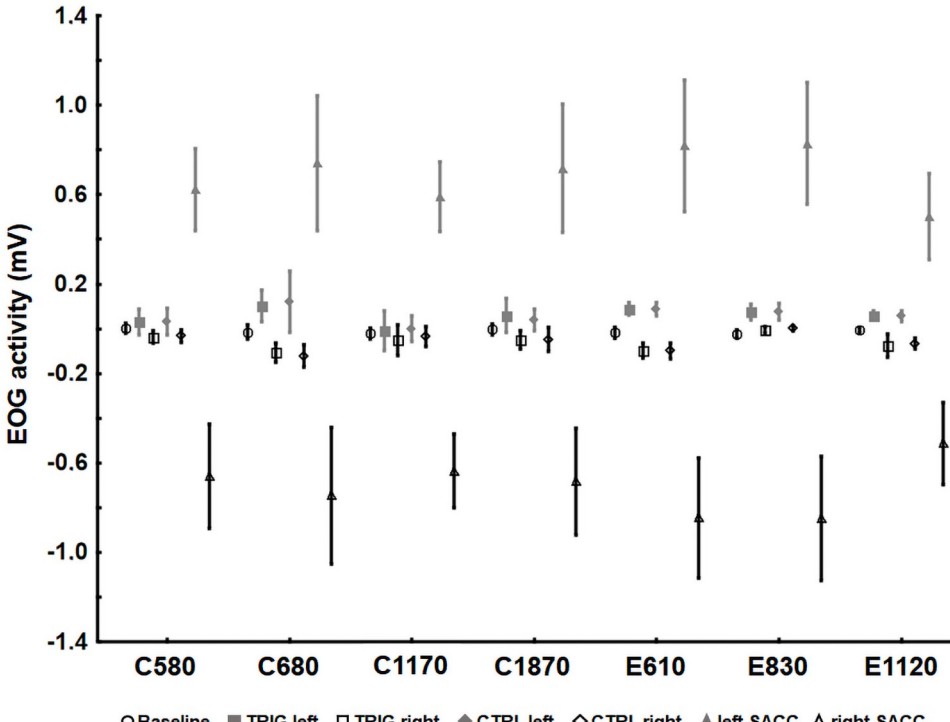

**Fig 5. Mean EOG activities calculated for the seven studies during baselines (circle) and cue-target periods in the cross-modal spatial cueing task, as a function of the cue type (square for TRIG and lozenge for CTRL) and the cue side (left in black or right in grey).** Mean EOG activities calculated during deliberate saccades (triangle) pointing to left or right targets during the calibration task. Error bars represent ± 1 standard error of the mean and are corrected for within-subject design [58]. CTRL = control trials, TRIG = trigeminal trials, SACC = saccade.

activity in all studies (all $p_s$ >.13) and were statistically different than during the voluntary saccades toward the target positions (all $p_s$ < .001). This result indicates that trigeminal cues did not trigger any saccade toward the target location during the cue-target period of the cross-modal spatial cueing task.

## Discussion

In this research, we tested whether nasal olfactory-trigeminal stimuli can exogenously capture visual attention. To do so, we used olfactory-trigeminal stimuli as lateralized cues and visual stimuli as lateralized targets in a cross-modal spatial cueing paradigm [29]. We recorded EOG activity in order to rule out the potential role of eye movements in this task. Moreover, we carefully considered the temporal window(s) in which this cross-modal attentional capture may occur. More specifically, we used different SOAs, SOA being a parameter known to be crucial in spatial cueing paradigms [32, 42, 59], and a potential reason why a previous study [36] failed to observe an attentional capture by olfactory-trigeminal stimuli on reaction times. We used seven SOAs, varying from (in average) 580 to 1870 ms. We found that olfactory-trigeminal stimuli can exogenously capture visual attention, as reflected by smaller RTs to valid compared to all other trials. This visual attention capture occurred for both stimuli we used: $CO_2$ and eucalyptol. However, this attentional capture was only present in specific temporal windows, i.e. with specific SOAs.

CO$_2$ did not capture visual attention when using a SOA of around 580 ms. These results complement Wudarczyk et al. [36]'s experiment, where the authors used a 500 ms SOA.

Congruently with our results, Wudarczyk et al. did not find an attentional capture by trigeminal stimuli, as reflected by RTs. With longer SOAs (i.e., 680 ms and 1170 ms in average), we found that reaction times in response to valid trials were about 20 ms shorter than for invalid trials (cue facilitating effect). These two studies provide unequivocal evidence of a visual attention capture by $CO_2$.

$CO_2$ almost exclusively stimulates the trigeminal system [20]. To gain ecological validity, we replicated our experimental procedure in three more experiments with eucalyptol as olfactory-trigeminal cue. Eucalyptol captured visual attention with a SOA of 610 ms in average. With SOAs of 830 ms and 1120 ms in average, this visual attention capture was no longer present.

It is tempting to speculate on the similarities and differences we found between these two trigeminal substances with different SOAs. For instance, it is possible that some trigeminal stimuli may capture attention faster and/or longer than others. Comparing systematically 2 of them–e.g., $CO_2$ and eucalyptol–would require using several and symmetrical SOAs for both. Moreover, if one wants to determine when attentional capture starts occurring and when it stops with these stimuli, much less wide time windows than the ones used here would be necessary. Consequently, rather than comparing the results we obtained with different SOAs and different stimuli, we prefer to emphasize the importance of carefully considering the SOA when studying attentional capture in a spatial cueing paradigm. This is true for trigeminal stimuli, but also for stimuli from other sensory modalities [30].

What we do know is that the time window where we found an attentional capture is congruent with the timeframe of a breath in [60]. Remember nevertheless that we chose to ask participants to refrain from breathing during a trial. This methodological choice was motivated by potential unwanted interactions between the natural flow of air and the one sent by the olfactometer. Although likely technically challenging, we encourage researchers to replicate our results while participants are naturally breathing. This would provide a more ecological protocol to study visual attention capture by trigeminal stimuli.

When the visual and auditory modality are involved, cross-modal attentional capture effects are already observed for SOAs of 200 ms (e.g., [61]). This is related to the high speed of stimulus processing for both modalities. For olfaction, trigeminal cue processing is much longer as compared to other sensory modalities [40–42, 62]. Early olfactory trigeminal event related potentials that reflect endogenous olfactory processes (i.e., N1, P1 and P2) are observed from 200 to 650 ms [20, 63, 64]. We speculate that there could not be any effect of trigeminal stimulation on the attention allocated to process the target until the perceptual processes of the cue are completed. This could explain why longer SOAs are needed in our paradigm to observe attentional capture by trigeminal stimuli. On the contrary, when the target appears after a critical period of time (i.e., long SOAs) we no longer observed evidence of an attentional capture by $CO_2$ or eucalyptol. In the same way that capture is observed for longer SOAs in olfactory than in visual, inhibition of return would be observed for longer SOAs in olfactory than in visual. We can speculate that we did not use SOAs long enough to observe this mechanism.

Contrary to Wudarczyk et al. [36], we did not find evidence of significant differences in accuracy between valid and invalid trials, neither in the first part or not the second part of our experiments (results not reported here). In Wudarczyk et al. [36]'s first experiment with $CO_2$ cues, there was a significant difference in accuracy in the 8 last trials but not in the 8 first ones. This difference was not associated with RTs differences between valid and invalid trials. As we had speculated, it is highly likely that the results we obtained here in terms of reaction times were due to the choice of SOA.

The extent to which the size of the facilitating effect we reported here is congruent with what is known for other cross-modal spatial cueing tasks is beyond the scope of this research

(particularly since too little olfactory cross-modal cueing data is available). The cueing effects we reported of about 25 ms for $CO_2$ and 40 ms for eucalyptol are close to those reported for other cross-modal experiments [26].

Hence, in more natural conditions, the ability to transfer spatial attention from trigeminal stimuli to visual ones gives a unique opportunity to exogenously orient attention towards purportedly relevant stimuli in the environment [21]. Congruently, in other sensory modalities, unpleasant [65], pleasant [24], and a-priori neutral stimuli associated with primary rewards [23, 66] can capture attention. In our experiments, most people considered the pure trigeminal stimulus (i.e., CO2) as unpleasant and the olfactory and trigeminal stimulus (i.e., eucalyptol) as pleasant. It is difficult in our series of studies to know whether the unpleasant and pleasant aspects of the trigeminal stimuli participate in the attentional capture we observed. For this, it would be particularly interesting to deliver trigeminal stimuli in one nostril while non trigeminal ones would be delivered in the other nostril. These two stimuli could be pleasant or unpleasant, which would help to disentangle the attentional capture effects linked to the trigeminal aspect from those linked to the valence of the stimulation.

Stimulations of the trigeminal system evoke different types of sensations: cooling, warmth, tingling or even pain [19, 20]. Theses sensations are caused by the activation of different types of trigeminal receptors (see [20] for a recent review). Here, $CO_2$ and eucalyptol molecules do not evoke the whole spectrum of trigeminal sensations. Using other molecules, future studies could investigate how the attentional capture highlighted in this research is specific to a type of receptors and/or sensations.

Only an experiment with variable SOAs and constant cue duration could formally answer the question of whether the different durations of trigeminal/olfactory-trigeminal cues have influenced the results. However, another important characteristic of the trigeminal system's functioning goes against the idea that habituation could explain the results we observed: repeated stimulation with high concentration of CO2 can activate pain fibers of the trigeminal nerve and even produce an increase in perceived intensity [67]. Longer SOAs should allow for more influence of trigeminal stimulation, thus potentially favouring more attentional capture or feedback inhibition. This is not what we observed–in our results, there seems to be an optimal SOA time window to observe attentional capture.

Finally, we want to point out an important limitation of this series of experiments, i.e. the lack of systematisation in the choice of the duration of the SOA. Its main objective was to find an SOA duration that could allow the observation of an attentional capture effect. It would have surely been more rigorous to fix a particular SOA (e.g., 1000 ms) and to increase or decrease it in steps of 250 ms for example. Moreover, this would have made possible to really observe the kinetics of the attentional capture effect. We can only hope that the flaw will be addressed in future research and will motivate researchers to study more rigorously the influence of SOA on the size of the attentional capture effect.

## Conclusion

Using a visual spatial cueing paradigm, we tested whether different trigeminal stimuli can capture visual attention. In seven experiments, we found robust evidence of an exogenous capture of visual attention by both $CO_2$ and eucalyptol. Together with Wudarczyk et al. [36]'s results, this data suggests not only that humans can detect, to some degree, the spatial location of chemical stimuli but also that this information is used to orient the spatial attention across modalities. In future studies examining the dynamics of this cross-modal attentional capture, we invite researchers to carefully choose the interval between the trigeminal cues and the visual targets: our data suggests that this attentional capture is present only in specific intervals.

## Acknowledgments

The authors thank Nadine Gaudreau and all the members of the Perception and Bioresponses Department of the Research and Development Division of Firmenich, SA, for their precious advice and their theoretical and technical competence. This study was conducted at the Brain and Behavior Laboratory (BBL) at the University of Geneva and benefited from the support of the BBL technical staff.

## Author Contributions

**Conceptualization:** Matthieu Ischer, David Sander, Sylvain Delplanque.

**Data curation:** Matthieu Ischer, Axel De Marles, Myriam Essellier.

**Formal analysis:** Matthieu Ischer, Sylvain Delplanque.

**Methodology:** Matthieu Ischer, Christelle Porcherot, Isabelle Cayeux, Christian Margot, Sylvain Delplanque.

**Project administration:** Sylvain Delplanque.

**Resources:** Christelle Porcherot, Isabelle Cayeux, Christian Margot.

**Writing – original draft:** Matthieu Ischer, Géraldine Coppin.

**Writing – review & editing:** Géraldine Coppin, David Sander, Sylvain Delplanque.

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
