## [Decision Letter · Decision Letter 0]

19 Jan 2021

PONE-D-20-35636

Automatic capture of visual spatial attention by olfactory trigeminal stimuli

PLOS ONE

Dear Dr. Delplanque,

Thank you for submitting your manuscript to PLOS ONE. After careful consideration, we feel that it has merit but does not fully meet PLOS ONE’s publication criteria as it currently stands. Therefore, we invite you to submit a revised version of the manuscript that addresses the points raised during the review process.

You will find the reviews of three reviewers and I recommend to follow their advice, especially Reviewer 1 has a number of comments that should be addressed.

We look forward to receiving your revised manuscript.

Kind regards,

Jessica Freiherr

Academic Editor

PLOS ONE

Journal Requirements:

2. Please ensure that you have specified how participants were assigned to the experiments described in the Methods section.

3.Thank you for stating the following in the Acknowledgments Section of your manuscript:

"This research was supported by the National Center of Competence in Research (NCCR) for the Affective Sciences, financed by a grant from the Swiss National Science Foundation hosted by the University of Geneva, and was also supported by a research grant from Firmenich, SA, to David Sander and Patrik Vuilleumier. The funders had no role in study design, data collection and analysis, decision to publish, or preparation of the manuscript."

b) We note that one or more of the authors is affiliated with the funding organization, indicating the funder may have had some role in the design, data collection, analysis or preparation of your manuscript for publication; in other words, the funder played an indirect role through the participation of the co-authors.

If the funding organization did not play a role in the study design, data collection and analysis, decision to publish, or preparation of the manuscript and only provided financial support in the form of authors' salaries and/or research materials, please review your statements relating to the author contributions, and ensure you have specifically and accurately indicated the role(s) that these authors had in your study in the Author Contributions section of the online submission form. Please make any necessary amendments directly within this section of the online submission form.  Please also update your Funding Statement to include the following statement: “The funder provided support in the form of salaries for authors [insert relevant initials], but did not have any additional role in the study design, data collection and analysis, decision to publish, or preparation of the manuscript. The specific roles of these authors are articulated in the ‘author contributions’ section.”

If the funding organization did have an additional role, please state and explain that role within your Funding Statement.

c) Please also provide an updated Competing Interests Statement declaring this commercial affiliation along with any other relevant declarations relating to employment, consultancy, patents, products in development, or marketed products, etc.  

Reviewers' comments:

Reviewer's Responses to Questions

**Comments to the Author**

1. Is the manuscript technically sound, and do the data support the conclusions?

Reviewer #1: Yes

Reviewer #2: Yes

Reviewer #3: Yes

2. Has the statistical analysis been performed appropriately and rigorously? 

Reviewer #1: Yes

Reviewer #2: Yes

Reviewer #3: Yes

3. Have the authors made all data underlying the findings in their manuscript fully available?

Reviewer #1: Yes

Reviewer #2: Yes

Reviewer #3: Yes

4. Is the manuscript presented in an intelligible fashion and written in standard English?

Reviewer #1: Yes

Reviewer #2: Yes

Reviewer #3: Yes

5. Review Comments to the Author

Reviewer #1: Ischer and colleagues present an interesting study consisting of seven experiments where the stimulus-onset asynchrony in a cross-modal chemosensory-visual cueing paradigm was systematically altered. Their results suggest that a pure trigeminal and a mixed olfactory-trigeminal stimulus are able to affect visuospatial attention in an exogeneous spatial cueing paradigm within specific time windows. While the research question is of interest, the sample size decent, and methodological approach sound and the interpretation adequate, I have several comments that I’d like to see addressed in a revised version of the manuscript.

Major:

• Research aim: The use of a mixed olfactory-trigeminal stimulus besides a pure trigeminal stimulus needs further justification. What is your hypothesis here? In the discussion, you acknowledge that you did not perform a formal and systematic comparison. Also, SOAs were different for both types. Why? (This needs to be introduced earlier to not leave the reader wondering about the purpose of this experimental manipulation.)

• The authors compute two levels of planned contrasts: 1) cue valid trials vs. all other trials (control valid, control invalid, cue invalid), and 2) cue valid vs. cue invalid. The second comparison should be motivated more explicitly. What is the expectation here? Did you choose this contrast because you assume a cost effect of the invalid cue (and if so, why not analyzing the effect of invalid cues in parallel to the effect of valid cues)? Why can the valid-invalid difference be regarded as the “size of the attentional capture effect”?

• Could the different durations (besides different onsets) of trigeminal/olfactory-trigeminal cues have influenced the results (e.g. due to habituation or inhibition of return)?

• The distribution of lateralization accuracy of control stimuli (Fig. 3B) suggests that the majority of subjects systematically indicated the opposite nostril (given that chance-performance would result in 5 correctly and 5 incorrectly classified trials). This would suggest a systematic bias – can you comment/clarify?

• While the authors demonstrate a facilitation effect of valid cues in specific time windows, there does not seem to be any inhibition of return for longer time windows. How can this be reconciled with the literature on cross-modal spatial cueing?

• Given accumulating evidence on sex-specific differences in chemosensory processing, it would be interesting to test the effect of sex on the spatial cueing effect (e.g., see Stuck et al., 2006, Clinical Neuropsychology, 117, 1367-1375).

Minor:

• Abstract: The statement about stimulus choice: “because they are supposedly relevant for the individuals” remains vague/unclear to the reader. I assume that the authors want to point out that trigeminal stimuli per se are salient stimuli for humans?

• I assume that the claim in line 87f “[…] but do not provide a formal demonstration of an automatic capture of visual attention” refers to the lack of an effect in reaction time? Please specify this central aspect, which seems to be the motivation of your study; the same is true for your first research aim: how do you define “automatic orientation of visual spatial attention” (especially in comparison to previous studies)?

• The authors are inconsistent in labeling the stimulus modality ( e.g. they use “olfactory” when supposedly referring to trigeminal or olfactory-trigeminal stimulus, e.g. in line 68); since the olfactory contribution to the exogeneous cueing effect in the eucalyptol condition is not clear, I recommend to strictly differentiate between both stimuli and use the descriptive labels “trigeminal” (CO2) and “olfactory-trigeminal” (eucalyptol) throughout the manuscript

• Sample: Were these samples independent, or did some subjects participate in several sub-studies? (In the extreme case, the majority of subjects participated in all experiments, which would suggest a different statistical approach.)

• Figure 1: Can you specify the meaning of different tube colors? (blue vs. black)

• I only realized in the results part that you also report significance (besides effect sizes, which I very much appreciate) for the planned contrast comparisons. Please add this information in the methods section as well.

• Were all statistical analyses performed with ESCI? If not, please specify the software used for these analyses.

• Fig 3 B: please specify in the figure legend that “accuracy” refers to “number of correct trials” (one often expects accuracy to be given as percent correct)

• Table 3: please highlight significant effects

• The efficiency score needs to be introduced in the methods section.

• Minor: language (e.g. line 80: being  be, line 205: saccade  saccades, line 238: consists in  consists of, line 297: in the air, line 348: in average  on average, line 367: can’t  cannot; line 430: in (60))

Reviewer #2: The studies reported in the manuscript by Delplanque and colleagues aim to investigate the automatic capture of visual spatial attention by trigeminal stimuli. The topic, cross-modal attentional capture with visual stimuli being cued by trigeminal stimulation, is, as the authors note, not well understood. Noteworthy, the authors’ convincing arguments on the importance of the considerations on the dynamics of this cross-modal attentional capture and thus, a careful selection of SOA, are of great significance to future studies in the field. The manuscript is well written and has a clear and easy-to-follow structure despite high information density. However, I have some concerns regarding the authors’ statements that sometimes appear speculative in nature. My comments mixing major and minor issues are listed below. I hope these points will be of help to the authors and the editors.

Abstract

• “We chose these stimuli because they are supposedly relevant for the individual, thus prone to capture attention” - this sentence does not fully fit into the context given by the following statement in the discussion: “However, the methodology used here does not allow any conclusions regarding the purportedly relevance of trigeminal stimuli, or whether the reported effects are linked to it”. It might be beneficial to specify in what context the stimuli are of relevance, given that they were rated as rather pleasant and thus not “alerting” in nature.

Methods

• Is there any reasoning behind the selected SOA values? If would be relevant to include.

Results

• Table 3 contains a typo (“C 1860” instead of “C 1870”).

• “Inverse efficiency scores” in C1170 and E610 were significantly smaller compared to all other trials. This statement is not complete since it is unclear whether smaller values mean higher or lower efficiency.

Discussion

• “Consistently with CO2 and a SOA around 680 ms, eucalyptol captured visual attention with a SOA of 610 ms in average.” This statement is confusing since it seems to neglect the fact that on contrary to the eucalyptol (E610), C580 did not capture visual attention although the SOA of 580ms is closer to 610ms than 680ms.

• “It is worth noting that the time window where we found evidence of an attentional

capture by both the CO2 and the eucalyptol matches the detection latencies (around 800 ms)”. Again, that statement seem to be inconclusive with the results since SOAs of around 800ms (E830 and E1120) were not of significance. Moreover, if the detection latency for both CO2 and eucalyptol is similar – one interpretation from the abovementioned sentence provided by the authors – it remains fully unexplained why the two stimuli show evidence of an attentional capture at different SOAs. Although the final conclusions the authors draw are rather cautious and clearly represent the limitations of the findings, the explanations of the observed effect provided in the discussion are partially of a speculative nature.

• The authors state that the odors they delivered to the participants were perceived as pleasant. If a rating of the stimuli was obtained it would be of interest to include it into the manuscript.

Reviewer #3: The overall aim of the paper is to show that trigeminal and bimodal olfactory stimuli can automatically capture humans’ spatial visual attention. Overall the manuscript is well written and presents interesting data that can certainly facilitate our understanding of attentional capture (by nasal trigeminal stimulations). However, I do have some concerns about the manner in which data has been presented in this manuscript.

1. Table 1. Identifying groups as C580, C680 is confusing.

2. Page 27 (line 463): consider rewriting the sentence starting with In the future,….

3. Page 27 (line 467). This paragraph is confusing. Consider rewriting or adding more explanations.

4. Is it possible to have a better Figure 3. Consider adding more explanation.

5. Page 26(line 436). Measuring SOA in olfaction is very difficult. The setup explaining the accurate measurement of SOA would be very critical for this manuscripts conclusions.

6. PLOS authors have the option to publish the peer review history of their article (what does this mean?). If published, this will include your full peer review and any attached files.

Reviewer #1: No

Reviewer #2: No

Reviewer #3: No

---

## [Author Response · Author response to Decision Letter 0]

25 Mar 2021

We have adapted the manuscript’ style following the recommendations.

2. Please ensure that you have specified how participants were assigned to the experiments described in the Methods section.

Number of participants (as well as gender, mean age ± SD) for the 7 experiments reported are presented in Table 1.

3.Thank you for stating the following in the Acknowledgments Section of your manuscript:

"This research was supported by the National Center of Competence in Research (NCCR) for the Affective Sciences, financed by a grant from the Swiss National Science Foundation hosted by the University of Geneva, and was also supported by a research grant from Firmenich, SA, to David Sander and Patrik Vuilleumier. The funders had no role in study design, data collection and analysis, decision to publish, or preparation of the manuscript."

Response: We have formulated the funding statement section in a separate paragraph as followed:

“This research was supported by the National Center of Competence in Research (NCCR) for the Affective Sciences, financed by a grant from the Swiss National Science Foundation hosted by the University of Geneva, and was also supported by a research grant from Firmenich, SA, to David Sander and Patrik Vuilleumier. The funder provided support in the form of salaries for author MI and provided material to conduct the experiments but did not have any additional role in the study design, data collection and analysis, decision to publish, or preparation of the manuscript.”

b) We note that one or more of the authors is affiliated with the funding organization, indicating the funder may have had some role in the design, data collection, analysis or preparation of your manuscript for publication; in other words, the funder played an indirect role through the participation of the co-authors.

If the funding organization did not play a role in the study design, data collection and analysis, decision to publish, or preparation of the manuscript and only provided financial support in the form of authors' salaries and/or research materials, please review your statements relating to the author contributions, and ensure you have specifically and accurately indicated the role(s) that these authors had in your study in the Author Contributions section of the online submission form. Please make any necessary amendments directly within this section of the online submission form. Please also update your Funding Statement to include the following statement: “The funder provided support in the form of salaries for authors [insert relevant initials], but did not have any additional role in the study design, data collection and analysis, decision to publish, or preparation of the manuscript. The specific roles of these authors are articulated in the ‘author contributions’ section.”

If the funding organization did have an additional role, please state and explain that role within your Funding Statement.

Response: The funding section now includes the following statement:

“[…] The funder provided support in the form of salaries for author MI and provided material to conduct the experiments but did not have any additional role in the study design, data collection and analysis, decision to publish, or preparation of the manuscript.”

c) Please also provide an updated Competing Interests Statement declaring this commercial affiliation along with any other relevant declarations relating to employment, consultancy, patents, products in development, or marketed products, etc. 

Response: We added a competing interests section that states:

“The authors declared no potential conflicts of interests with respect to the authorship and/or publication of this article. The commercial affiliation of CP, IC & CM does not alter our adherence to PLOS ONE policies on sharing data and materials.” 

Reviewer #1: Ischer and colleagues present an interesting study consisting of seven experiments where the stimulus-onset asynchrony in a cross-modal chemosensory-visual cueing paradigm was systematically altered. Their results suggest that a pure trigeminal and a mixed olfactory-trigeminal stimulus are able to affect visuospatial attention in an exogeneous spatial cueing paradigm within specific time windows. While the research question is of interest, the sample size decent, and methodological approach sound and the interpretation adequate, I have several comments that I’d like to see addressed in a revised version of the manuscript.

Major:

• Research aim: The use of a mixed olfactory-trigeminal stimulus besides a pure trigeminal stimulus needs further justification. What is your hypothesis here? In the discussion, you acknowledge that you did not perform a formal and systematic comparison.

Response: We thank the reviewer for this interesting point. In a first step, we used a purely trigeminal stimulation, which prevented us from introducing potential interactions between the trigeminal and the olfactory systems. This allowed us to put ourselves in the "ideal" experimental situation, i. e. restricting the observed effects exclusively to an activation of the trigeminal system. In a second step, we considered the "ecological" argument, i.e. that CO2 at the used concentrations is not a very common stimulation in nature. The phenomenon of attentional capture that we had observed with CO2 could then be artificial, which would have reduced the scope of the results. We therefore decided to use a mixed olfactory and trigeminal compound that is easily found in our daily lives in order to extend the conclusions brought by CO2. The trigeminal system is activated via several types of receptors (e.g., Frasnelli and Manescu, 2017) that are more or less specific to certain compounds. It seemed interesting to study attentional capture effects for at least two types of receptors/molecules to rule out the possibility that these attentional effects are solely and quite artificially produced by CO2. We hypothesized that as soon as a compound or gas has the property to stimulate the trigeminal system, we should observe an attentional capture, i.e., that reaction times of valid trial cued by pure trigeminal (CO2) and olfactory-trigeminal (eucalyptol) stimulations will be shorter than the reaction times for all other trials.

Frasnelli J, Manescu S. The Intranasal Trigeminal System. In: Buettner A, editor. Springer Handbook of Odor [Internet]. Cham: Springer International Publishing; 2017. p. 113–4. Available from: http://link.springer.com/10.1007/978-3-319-26932-0_46

Also, SOAs were different for both types. Why? (This needs to be introduced earlier to not leave the reader wondering about the purpose of this experimental manipulation.)

Response: The unorthodox choice of SOAs is undoubtedly the point that raises the most questions about this series of 7 experiences. We can only humbly admit that it is a lack of systematisation in the protocol design. The main objective of the doctoral thesis work that we present in this article was to find an SOA duration that could allow the observation of an attentional capture effect. We agree that it would have been more rigorous to fix a particular SOA (e.g., 1000 ms) and to increase or decrease it in steps of 250 ms for example. This would have made possible to really observe the kinetics of the attentional capture effect. Observing the size of the capture effect as a function of time and molecule is unfortunately a question that appeared too late in the eyes of the principal investigator of this research. These 7 experiments carried out within the framework of a doctoral thesis could not be completed/re-done to integrate this rigour. In spite of this flaw that future more systematic research may correct, we wanted to share with the community the result that attentional capture is possible through pure trigeminal or mixed stimuli. This result was only suggested by one study before, and not in terms of reaction times. We fully understand that this lack of systematisation leaves a bitter taste because the story would certainly have been more elegant if the SOAs had been better chosen. We cannot elaborate on post hoc reasons that would justify the durations of SOAs. That would be dishonest. We therefore deliver all the information in our possession with its strengths and weaknesses. We can only hope that the questions raised by this lack of systematisation will motivate researchers to study more rigorously the influence of SOA on the size of the attentional capture effect.

• The authors compute two levels of planned contrasts: 1) cue valid trials vs. all other trials (control valid, control invalid, cue invalid), and 2) cue valid vs. cue invalid. The second comparison should be motivated more explicitly. What is the expectation here? Did you choose this contrast because you assume a cost effect of the invalid cue (and if so, why not analyzing the effect of invalid cues in parallel to the effect of valid cues)? Why can the valid-invalid difference be regarded as the “size of the attentional capture effect”?

Response: We chose not to perform an Omnibus test with all the post hoc comparisons because we had very clear a priori hypotheses on what to expect on mean reaction times in which an attentional capture existed. We fully agree with the reviewer that the comparison invalid versus valid trigeminal stimuli is unnecessary and have removed it from the revised manuscript. We added the sizes of the differences in the manuscript to give the reader information on the magnitude of the effect. The reviewer also raised the very relevant question of a cost effect in the invalid trigeminal condition. We have therefore added a small note that presents the corresponding statistics that make this hypothesis highly unlikely.

“To examine possible cost effects that might have been observed for invalid trigeminal trials (attention retained at the invalid location), we performed another planned contrast comparison on correct reaction times (RT in ms). Contrast cell weights were as follows: −1 for the invalid air cell, −1 for the valid air cell, + 2 for the for the invalid trigeminal cell, and 0 for valid trigeminal cell. For the three experiments that revealed an attentional capture, response times did not statistically differ between trigeminal invalid trials as compared to all other control trials, all Fs < 1.84, ps > .18, η2s < .07.”

We would like to warmly thank the reviewer who, through the modifications we have made following his/her comments, has allowed for a much more direct and precise demonstration.

• Could the different durations (besides different onsets) of trigeminal/olfactory-trigeminal cues have influenced the results (e.g. due to habituation or inhibition of return)?

This is a very good question but we cannot answer it precisely without doing an experiment with a constant cue duration and variable SOAs. However, some knowledge of the trigeminal system’s functioning can provide some answers. It has been demonstrated that repeated stimulation with high concentration of CO2 may activate pain fibers of the trigeminal nerve and even produce an increase in perceived intensity (Hummel et al., 1994). This result goes against the idea that habituation could explain the results we observed. Longer SOAs should allow for more influence of trigeminal stimulation, thus potentially favouring more attentional capture or feedback inhibition. This is not what we observed, it looks like there is an optimal SOA time window to observe attentional capture. Our results did not allow the characterisation it in detail but the dynamics outlined could not be explained by habituation effects.

Hummel T, Gruber M, Pauli E, Kobal G. Chemo-somatosensory event-related potentials in response to repetitive painful chemical stimulation of the nasal mucosa. Electroencephalography and Clinical Neurophysiology/Evoked Potentials Section. 1994 Sep;92(5):426–32.

• The distribution of lateralization accuracy of control stimuli (Fig. 3B) suggests that the majority of subjects systematically indicated the opposite nostril (given that chance-performance would result in 5 correctly and 5 incorrectly classified trials). This would suggest a systematic bias – can you comment/clarify?

Response: Thanks to the reviewer for raising this point, which was unproperly described in the previous version of the manuscript. After re-checking the E-prime scripts, we can confirm that participants did not respond if they did not feel anything. Since the E-prime software has coded errors and non-responses in the same way, it is not possible to separate true errors from non-responses. This does not allow us to observe the pattern of responses that the reviewer had anticipated (chance performance). This now reads as:

“For each trial (every 15 s), participants were requested to indicate if they felt any change in the left or right nostril by pressing the left or right arrow of a keyboard, accordingly. They were requested not to press if they could not feel any change at all.” 

• While the authors demonstrate a facilitation effect of valid cues in specific time windows, there does not seem to be any inhibition of return for longer time windows. How can this be reconciled with the literature on cross-modal spatial cueing?

Response: Cross-modal attentional capture effects are already observed for SOAs of 200 ms (e.g., Sanz et al., 2018) and cross modal inhibition of return for SOAs around 800 ms (e.g., Wang et al., 2012) for vision and hearing. We have just cited these two references to illustrate that when the visual and auditory modality are involved, attentional capture and inhibition of return phenomena occur with short SOAs. This is related to the high speed of stimulus processing for both modalities. For olfaction, trigeminal cue processing is much longer. The first observable evoked potentials for attended trigeminal stimulation appear around 400 ms (e.g., Geisler et al., 2000) which is much later than for visual stimulation. This could explain why longer SOAs are needed in our paradigm to observe attentional capture by trigeminal stimuli. In the same way that capture is observed for longer SOAs in olfactory than in visual, inhibition of return would be observed for longer SOAs in olfactory than in visual. We can speculate that we did not use SOAs long enough to observe this mechanism, if it exists. 

Geisler MW, Murphy C. Event-related brain potentials to attended and ignored olfactory and trigeminal stimuli. International Journal of Psychophysiology. 2000 Sep;37(3):309–15. 

Sanz LRD, Vuilleumier P, Bourgeois A. Cross-modal integration during value-driven attentional capture. Neuropsychologia. 2018 Nov;120:105–12.

Wang L, Yue Z, Chen Q. Cross-modal nonspatial repetition inhibition. Atten Percept Psychophys. 2012 Jul;74(5):867–78.

• Given accumulating evidence on sex-specific differences in chemosensory processing, it would be interesting to test the effect of sex on the spatial cueing effect (e.g., see Stuck et al., 2006, Clinical Neuropsychology, 117, 1367-1375).

Response: We could not agree more with the reviewer, men and women differ in many aspects of olfactory processes. The work cited by the reviewer points to differences in lateralization performance that would be superior in women. As a result, attentional capture could be different in men and women. Given the insufficient number of men and women in each experiment to examine a gender effect, we pooled the three experiments that demonstrated attentional capture. We obtained a sample of 52 women and 29 men. The contrast analysis (weighted -1, -1, -1, 3 for within and -1, 1 for between contrasts) performed on this new matrix didn’t reveal a statistically significant difference, F(1,167) = 2.42, p = .12, η2 = .014, 90% CI = [0, .057]. As this paper did not set out to explore the influence of gender on attentional capture, we would prefer not to include this lack of significant result in the manuscript.

Minor:

• Abstract: The statement about stimulus choice: “because they are supposedly relevant for the individuals” remains vague/unclear to the reader. I assume that the authors want to point out that trigeminal stimuli per se are salient stimuli for humans?

Response: We are sorry for the lack of clarity in this statement. The molecules used here would be particularly salient/relevant for the individual because i) they activate the trigeminal system and ii) the trigeminal system is supposed to be an alarm system for the organism (as it conveys sensations of pain, cooling, warmth and intensity). We have clarified this point as follows:

« We chose these stimuli because they activate the trigeminal system which can be considered as an alert system and are thus supposedly relevant for the individual, and thus prone to capture attention.”

• I assume that the claim in line 87f “[…] but do not provide a formal demonstration of an automatic capture of visual attention” refers to the lack of an effect in reaction time? Please specify this central aspect, which seems to be the motivation of your study; 

Response: Attentional capture is demonstrated either by a decrease in reaction times or by an increase in accuracy in response to valid trials compared invalid trials. It is consequently not accurate to state that Wudarczyk and colleagues did not perform a formal demonstration of the attentional capture because they did not observe any results on reaction times. They have observed results in accuracy. We have therefore reformulated this point as follows:

« Using these cues and targets, Wudarczyk and colleagues (36) showed that, although participants were not statistically faster for valid trials than for invalid ones, they were more accurate in the valid condition than in the invalid one. These results consequently suggest that lateralized trigeminal cues can influence visual spatial attention. We believe that the demonstration of an exogenous capture of visual attention by trigeminal stimulation on reaction times could be provided by changing one critical aspect of the experimental design: the Stimulus Onset Asynchrony (SOA).”

[…] the same is true for your first research aim: how do you define “automatic orientation of visual spatial attention” (especially in comparison to previous studies)?

This very important remark by the reviewer made us give up using the term "automatic", considering the great difficulty of formally establishing automaticity (Santangelo and Spence, 2008). We cannot assert automaticity with our protocol, but the term exogenous is much more rigorous and corresponds perfectly to our situation.

Santangelo V, Spence C. Is the exogenous orienting of spatial attention truly automatic? Evidence from unimodal and multisensory studies. Consciousness and Cognition. 2008 Sep;17(3):989–1015.

We changed the title accordingly.

• The authors are inconsistent in labeling the stimulus modality ( e.g. they use “olfactory” when supposedly referring to trigeminal or olfactory-trigeminal stimulus, e.g. in line 68); since the olfactory contribution to the exogeneous cueing effect in the eucalyptol condition is not clear, I recommend to strictly differentiate between both stimuli and use the descriptive labels “trigeminal” (CO2) and “olfactory-trigeminal” (eucalyptol) throughout the manuscript

Response: We thank the reviewer for this recommendation which we have adopted throughout the manuscript.

• Sample: Were these samples independent, or did some subjects participate in several sub-studies? (In the extreme case, the majority of subjects participated in all experiments, which would suggest a different statistical approach.)

Response: Participants took part in only one experiment. The 7 samples were consequently independent.

• Figure 1: Can you specify the meaning of different tube colors? (blue vs. black)

Response: The colour of the tubes differentiates the part that carries air (control condition) from the part that carries trigeminal stimuli.

• I only realized in the results part that you also report significance (besides effect sizes, which I very much appreciate) for the planned contrast comparisons. Please add this information in the methods section as well.

Response: We have added this information.

• Were all statistical analyses performed with ESCI? If not, please specify the software used for these analyses.

Response: Statistical analyses were carried out using Statistica software (now mentioned in the methods section). ESCI software was used only for the valid vs. invalid trigeminal comparison. As this comparison is no longer in the manuscript, we have removed this mention.

• Fig 3 B: please specify in the figure legend that “accuracy” refers to “number of correct trials” (one often expects accuracy to be given as percent correct)

Response: Following this comment as well as a request from another reviewer, we have completely reworked Figure 3 to make it more explicit.

Figure 3: A) Boxplots and violin plots of CO2 concentrations (% in air) used for the spatial cueing tasks with CO2 as a function of SOA (in ms). B) Mean correct responses (± SEM) to the lateralization tasks for all types of stimulations, for all studies. TRIG = trigeminal trials, CTRL = control trials (clean air), EXPE = experiment name. C) Mean correct responses (± SEM) to the different conditions of the cross-modal spatial cueing tasks, for CO2 and Eucalyptol studies, separately. TRIG = trigeminal trials, CTRL = control trials, i = invalid trials and v = valid trials.

• Table 3: please highlight significant effects

Response: In order to be consistent with the statistics presented in the text, we have modified Table 3 to present the means and SEMs of the reaction times for the valid trigeminal trials vs. the mean of all other trials. Researchers who wish to obtain the averages of reaction times for each separate condition can easily use the raw data that we share without restriction. In addition, statistically significant means are now presented in bold italics.

Table 3: Mean (M) reaction times (± SEM) for the trigeminal valid (TRIG valid) and all other conditions together (Others) obtained during the cross-modal spatial cueing tasks for all studies. Statistically significant different means are presented in bold italics.

• The efficiency score needs to be introduced in the methods section.

Response: We made this change.

• Minor: language (e.g. line 80: being  be, line 205: saccade  saccades, line 238: consists in  consists of, line 297: in the air, line 348: in average  on average, line 367: can’t  cannot; line 430: in (60))

Response: We made all the changes, thanks a lot.

Reviewer #2: The studies reported in the manuscript by Delplanque and colleagues aim to investigate the automatic capture of visual spatial attention by trigeminal stimuli. The topic, cross-modal attentional capture with visual stimuli being cued by trigeminal stimulation, is, as the authors note, not well understood. Noteworthy, the authors’ convincing arguments on the importance of the considerations on the dynamics of this cross-modal attentional capture and thus, a careful selection of SOA, are of great significance to future studies in the field. The manuscript is well written and has a clear and easy-to-follow structure despite high information density. However, I have some concerns regarding the authors’ statements that sometimes appear speculative in nature. My comments mixing major and minor issues are listed below. I hope these points will be of help to the authors and the editors.

Abstract

• “We chose these stimuli because they are supposedly relevant for the individual, thus prone to capture attention” - this sentence does not fully fit into the context given by the following statement in the discussion: “However, the methodology used here does not allow any conclusions regarding the purportedly relevance of trigeminal stimuli, or whether the reported effects are linked to it”. It might be beneficial to specify in what context the stimuli are of relevance, given that they were rated as rather pleasant and thus not “alerting” in nature.

Response: As also raised by the first reviewer, we have indeed not been clear enough in this formulation in the abstract. We choose these two molecules because they activate the trigeminal system and the latter is supposed to be an alarm system for the organism (because it conveys sensations of pain, cooling, warmth and intensity). This is the reason why these molecules would be particularly salient/relevant for the individual. We have clarified this point as follows:

« We chose these stimuli because they activate the trigeminal system which can be considered as an alert system and are thus supposedly relevant for the individual, and thus prone to capture attention.”

On the other hand, we agree with the reviewer that we failed to clearly deliver the idea we wanted to bring into the discussion. We have therefore reworded the two paragraphs that raised questions as follows:

“In our experiments, most people considered the pure trigeminal stimulus (i.e., CO2) as unpleasant and the olfactory and trigeminal stimulus (i.e., eucalyptol) as pleasant. It is difficult in our series of studies to know whether the unpleasant and pleasant aspects of the trigeminal stimuli participate in the attentional capture we observed. For this, it would be particularly interesting to deliver trigeminal stimuli in one nostril while non trigeminal ones would be delivered in the other nostril. These two stimuli could be pleasant or unpleasant, which would help to disentangle the attentional capture effects linked to the trigeminal aspect from those linked to the valence of the stimulation.”

Methods

• Is there any reasoning behind the selected SOA values? If would be relevant to include.

Response: As we mentioned in reviewer 1’ second comment, the unorthodox choice of SOAs is undoubtedly the point that raises the most questions about this series of 7 experiences. We can only humbly admit that it is a lack of systematisation in the design of the protocols. The main objective of the thesis work that we present in this article is to find an SOA duration that could allow the observation of an attentional capture effect. We agree that it would have been more rigorous to fix a particular SOA (e.g. 1000ms) and to increase or decrease it in steps of 250ms for example. This would have made it possible to really observe the kinetics of the attentional capture effect. Observing the size of the capture effect as a function of time and molecule is unfortunately a question that appeared too late in the eyes of the principal investigator of this research. These 7 experiments carried out within the framework of a doctoral thesis could not be completed/re-done to integrate this rigour. In spite of this flaw that future more systematic research may correct, we wanted to share with the community the result that attentive capture is possible through pure trigeminal or mixed stimuli. This result was only suggested by one study before, and not in terms of reaction times. We fully understand that this lack of systematisation leaves a bitter taste because the story would certainly have been more beautiful to tell if the SOAs had been better chosen. We cannot elaborate on post hoc reasons that would justify the durations of SOAs. That would be dishonest. We therefore deliver all the information in our possession with its strengths and weaknesses. We can only hope that the questions raised by this lack of systematisation will motivate researchers to study more rigorously the influence of SOA on the size of the attentional capture effect.

Results

• Table 3 contains a typo (“C 1860” instead of “C 1870”).

Response: Thank you for pointing this out, we made this change.

Table 3: Mean (M) reaction times (± SEM) for the trigeminal valid (TRIG valid) and all other conditions together (Others) obtained during the cross-modal spatial cueing tasks for all studies. Statistically significant different means are presented in bold italics.

Response: Please note that table 3 now contains only two conditions which are trigeminal valid stimuli on one side and all other stimuli on the other. The means thus presented best correspond to the statistical contrasts performed.

• “Inverse efficiency scores” in C1170 and E610 were significantly smaller compared to all other trials. This statement is not complete since it is unclear whether smaller values mean higher or lower efficiency.

Response: Thank you for pointing this out. We have now clarified what this decrease in "reverse efficiency" means.

“Inverse efficiency scores were smaller, indicating an increase in efficiency in response to valid trigeminal trials when compared to all other trials…”

Discussion

• “Consistently with CO2 and a SOA around 680 ms, eucalyptol captured visual attention with a SOA of 610 ms in average.” This statement is confusing since it seems to neglect the fact that on contrary to the eucalyptol (E610), C580 did not capture visual attention although the SOA of 580ms is closer to 610ms than 680ms.

Response: We rephrased this part:

« Eucalyptol captured visual attention with a SOA of 610 ms in average. With SOAs of 830 ms and 1120 ms in average, this visual attention capture was no longer present. »

• “It is worth noting that the time window where we found evidence of an attentional capture by both the CO2 and the eucalyptol matches the detection latencies (around 800 ms)”. Again, that statement seem to be inconclusive with the results since SOAs of around 800ms (E830 and E1120) were not of significance. Moreover, if the detection latency for both CO2 and eucalyptol is similar – one interpretation from the abovementioned sentence provided by the authors – it remains fully unexplained why the two stimuli show evidence of an attentional capture at different SOAs. Although the final conclusions the authors draw are rather cautious and clearly represent the limitations of the findings, the explanations of the observed effect provided in the discussion are partially of a speculative nature.

Response: It is true that the lack of systematization in the choice of SOAs does not allow us to discuss the precise timing of the appearance and disappearance of these attentional capture effects. We were therefore quite speculative and we thank the reviewer for pointing this out. However, we can discuss the much more robust evidence that the SOAs for which effects are observed are longer in our paradigm than in cross-modal paradigms involving visual or auditory stimuli. We have reformulated our discussion on this point by focusing on the comparison with the other sensory modalities :

« When the visual and auditory modality are involved, cross-modal attentional capture effects are already observed for SOAs of 200 ms (e.g., Sanz et al., 2018). This is related to the high speed of stimulus processing for both modalities. For olfaction, trigeminal cue processing is much longer as compared to other sensory modalities (40–42,61). Early olfactory trigeminal event related potentials that reflect perceptual olfactory processes (i.e., N1, P1 and P2) are observed from 200 to 650 ms (20,62,63). We speculate that there could not be any effect of trigeminal stimulation on the attention allocated to process the target until the perceptual processes of the cue are completed. This could explain why longer SOAs are needed in our paradigm to observe attentional capture by trigeminal stimuli. On the contrary, when the target appears after a critical period of time (i.e., long SOAs) we no longer observed evidence of an attentional capture by CO2 or eucalyptol. »

• The authors state that the odors they delivered to the participants were perceived as pleasant. If a rating of the stimuli was obtained it would be of interest to include it into the manuscript.

Response: Unfortunately, we did not formally test the valence of Eucalyptol. We just asked the participants if they liked the smell and they all liked it. Moreover, they were informed during recruitment before starting the study that they were going to smell of eucalyptol/eucalyptus, so the participants who did not like it did not come to participate.

Reviewer #3: The overall aim of the paper is to show that trigeminal and bimodal olfactory stimuli can automatically capture humans’ spatial visual attention. Overall the manuscript is well written and presents interesting data that can certainly facilitate our understanding of attentional capture (by nasal trigeminal stimulations). However, I do have some concerns about the manner in which data has been presented in this manuscript.

We thank reviewer 3 for these encouraging comments.

1. Table 1. Identifying groups as C580, C680 is confusing.

Response: We have reworked Table 1 to clarify what the names of the experiments mean.

2. Page 27 (line 463): consider rewriting the sentence starting with In the future,….

AND 3. Page 27 (line 467). This paragraph is confusing. Consider rewriting or adding more explanations.

Response: Thank you for pointing the lack of clarity in these two paragraphs. We have reworded them as follows:

“In our experiments, most people considered the pure trigeminal stimulus (i.e., CO2) as unpleasant and the olfactory and trigeminal stimulus (i.e., eucalyptol) as pleasant. It is difficult in our series of studies to know whether the unpleasant and pleasant aspects of the trigeminal stimuli participate in the attentional capture we observed. For this, it would be particularly interesting to deliver trigeminal stimuli in one nostril while non trigeminal ones would be delivered in the other nostril. These two stimuli could be pleasant or unpleasant, which would help to disentangle the attentional capture effects linked to the trigeminal aspect from those linked to the valence of the stimulation.”

4. Is it possible to have a better Figure 3. Consider adding more explanation.

Response: We have completely reworked Figure 3 to make it more explicit. 

Figure 3: A) Boxplots and violin plots of CO2 concentrations (% in air) used for the spatial cueing tasks with CO2 as a function of SOA (in ms). B) Mean correct responses (± SEM) to the lateralization tasks for all types of stimulations, for all studies. TRIG = trigeminal trials, CTRL = control trials (clean air), EXPE = experiment name. C) Mean correct responses (± SEM) to the different conditions of the cross-modal spatial cueing tasks, for CO2 and Eucalyptol studies, separately. TRIG = trigeminal trials, CTRL = control trials, i = invalid trials and v = valid trials.

5. Page 26(line 436). Measuring SOA in olfaction is very difficult. The setup explaining the accurate measurement of SOA would be very critical for this manuscripts conclusions.

Response: We fully agree that accurately calculating the time between the rapid presentation of two odors is a difficult task. In our experiments, it is important to know the time between the presentation of the gas or molecule (as soon as the olfactometer valve opens) and the presentation of the visual target on the screen. Calculating the time between the onset of the odor and the onset of a picture is much easier, since the end of the gas or molecule presentation is less critical. The SOAs reported in this article were obtained from the E-prime software which provides one output file per participant for all experiments. It is this same software that controls the opening and closing of the olfactometer valves and the display of targets on the computer screen. So we have all the information about the different timing of the opening of the valves and the appearance of the targets for each trial, participant and experiment. The SOA is given by subtracting these two values. Moreover, for each experiment we also performed a check that the trigeminal stimuli were delivered during the SOA. It is not possible to perform this check for each participant because one cannot both perform the experiment and check for stimulus delivery with an external device. In an independent run, we connected the end of the tubes to an analogue mass flow meter (Bronkhorst, MASS-VIEW® MV-102) connected to the Biopac MP150 data acquisition system. We then recorded the flow of CO2 or air containing eucalyptol for the different trials and experiments to confirm that the gas or molecule was delivered during the SOA. An example of a recording for a particular test is shown below. 

We can therefore confirm that the stimuli were delivered during the SOA calculated from the timings provided by E-Prime. The individual data of the SOAs are available at:

https://osf.io/6vpzh/?view_only=ef734433920c41bb84da239ed798f8db#show_login.

We thank reviewer 3 for his/her comments.

---

## [Decision Letter · Decision Letter 1]

14 May 2021

PONE-D-20-35636R1

Exogenous capture of visual spatial attention by olfactory trigeminal stimuli

PLOS ONE

Dear Dr. Delplanque,

Thank you for submitting your manuscript to PLOS ONE. After careful consideration, we feel that it has merit but does not fully meet PLOS ONE’s publication criteria as it currently stands. Therefore, we invite you to submit a revised version of the manuscript that addresses the points raised during the review process.

We look forward to receiving your revised manuscript.

Kind regards,

Jessica Freiherr

Academic Editor

PLOS ONE

Journal Requirements:

Additional Editor Comments (if provided):

Reviewer 1 still asks for some more details and it would be great if you could accomplish that. Thanks a lot.

Reviewers' comments:

Reviewer's Responses to Questions

**Comments to the Author**

1. If the authors have adequately addressed your comments raised in a previous round of review and you feel that this manuscript is now acceptable for publication, you may indicate that here to bypass the “Comments to the Author” section, enter your conflict of interest statement in the “Confidential to Editor” section, and submit your "Accept" recommendation.

Reviewer #1: (No Response)

Reviewer #2: All comments have been addressed

2. Is the manuscript technically sound, and do the data support the conclusions?

Reviewer #1: Yes

Reviewer #2: Yes

3. Has the statistical analysis been performed appropriately and rigorously? 

Reviewer #1: Yes

Reviewer #2: Yes

4. Have the authors made all data underlying the findings in their manuscript fully available?

Reviewer #1: Yes

Reviewer #2: (No Response)

5. Is the manuscript presented in an intelligible fashion and written in standard English?

Reviewer #1: Yes

Reviewer #2: Yes

6. Review Comments to the Author

Reviewer #1: I’m glad to see that my comments have been addressed thoroughly by the authors. Some clarifications were, however, not incorporated into the manuscript. As a minor revision, I encourage the authors to do this since other readers might rise these questions as well, at least for comments #1 (research aim), #2 (different SOAs; I appreciate the honest answer here and suggest to address this fact as a limitation), and #4 (effect of different cue durations).

Reviewer #2: (No Response)

7. PLOS authors have the option to publish the peer review history of their article (what does this mean?). If published, this will include your full peer review and any attached files.

Reviewer #1: No

Reviewer #2: No

---

## [Author Response · Author response to Decision Letter 1]

17 May 2021

Dear Dr. Jessica Freiherr,

We would like to thank the reviewers and you for your constructive feedback about our manuscript “Exogenous capture of visual spatial attention by olfactory trigeminal stimuli” (Manuscript PONE-D-20-35636), and for giving us the opportunity to improve it further in this revised version. We have carefully addressed the minor points that reviewer 1 has raised (paragraphs in bold in this letter) and revised our manuscript accordingly, as detailed below. 

Attached to this revision letter, we have included a marked-up copy of the changes made to the previous Manuscript, using Microsoft word’s Track Changes function to make the changes transparent.

We thank you and the reviewers for your time and consideration.

With our best regards,

Sylvain Delplanque

On behalf of co-authors

Reviewer #1: I’m glad to see that my comments have been addressed thoroughly by the authors. Some clarifications were, however, not incorporated into the manuscript. As a minor revision, I encourage the authors to do this since other readers might rise these questions as well, at least for comments #1 (research aim), #2 (different SOAs; I appreciate the honest answer here and suggest to address this fact as a limitation), and #4 (effect of different cue durations).

We thank reviewer 1 again for his/her suggestions.

We have included our responses to his/her comments in the current version of the manuscript.

Comment 1: Introduction, lines 96-118:

This research therefore aims at i) determining whether a lateralized olfactory-trigeminal stimulus exogenously orientates visual spatial attention and ii) providing a first indication on the time window in which this attentional capture occurs. To do so, the trigeminal system was stimulated with CO2 and eucalyptol. CO2 gas provokes stinging and/or pungency sensations in the nose (45). Since the CO2 almost exclusively stimulates the trigeminal system (20,46), it provides a rare and ideal candidate to investigate trigeminal interactions with visual spatial attention. Thus, in a first step, we used a purely trigeminal stimulation, which prevented us from introducing potential interactions between the trigeminal and the olfactory systems. This allowed us to put ourselves in the "ideal" experimental situation, i.e. restricting the observed effects exclusively to an activation of the trigeminal system. We also used eucalyptol, which is more likely to be found in natural situations. Eucalyptol activates both the olfactory and trigeminal system (20) and is widely used in clinical research to assess trigeminal sensitivity (15). Thus, in a second step, we considered the "ecological" argument, i.e. that CO2 at the used concentrations is not a very common stimulation in nature. Consequently, the phenomenon of attentional capture that we expected to observe with CO2 could be artificial, which would have reduced the scope of the results. We therefore decided to use a mixed olfactory and trigeminal compound that is easily found in our daily lives in order to extend the conclusions brought by CO2. The trigeminal system is activated via several types of receptors (e.g.,20) that are more or less specific to certain compounds. It seemed interesting to study attentional capture effects for at least two types of receptors/molecules to rule out the possibility that these attentional effects are solely and quite artificially produced by CO2. We hypothesized that as soon as a compound or gas has the property to stimulate the trigeminal system, we should observe an attentional capture, i.e., that reaction times of valid trial cued by pure trigeminal (CO2) and olfactory-trigeminal (eucalyptol) stimulations will be shorter than the reaction times for all other trials.

Comment 2: Discussion, lines 451-458:

Only an experiment with variable SOAs and constant cue duration could formally answer the question of whether the different durations of trigeminal/olfactory-trigeminal cues have influenced the results. However, another important characteristic of the trigeminal system’s functioning goes against the idea that habituation could explain the results we observed: repeated stimulation with high concentration of CO2 can activate pain fibers of the trigeminal nerve and even produce an increase in perceived intensity (66). Longer SOAs should allow for more influence of trigeminal stimulation, thus potentially favouring more attentional capture or feedback inhibition. This is not what we observed – in our results, there seems to be an optimal SOA time window to observe attentional capture.

Comment 4: Discussion, lines 459-465:

Finally, we want to point out an important limitation of this series of experiments, i.e. the lack of systematisation in the choice of the duration of the SOA. Its main objective was to find an SOA duration that could allow the observation of an attentional capture effect. It would have surely been more rigorous to fix a particular SOA (e.g., 1000 ms) and to increase or decrease it in steps of 250 ms for example. Moreover, this would have made possible to really observe the kinetics of the attentional capture effect. We can only hope that the flaw will be addressed in future research and will motivate researchers to study more rigorously the influence of SOA on the size of the attentional capture effect.

---

## [Editor Report · Decision Letter 2]

26 May 2021

Exogenous capture of visual spatial attention by olfactory trigeminal stimuli

PONE-D-20-35636R2

Dear Dr. Delplanque,

We’re pleased to inform you that your manuscript has been judged scientifically suitable for publication and will be formally accepted for publication once it meets all outstanding technical requirements.

Kind regards,

Jessica Freiherr

Academic Editor

PLOS ONE
---

## [Editor Report · Acceptance letter]

1 Jun 2021

PONE-D-20-35636R2 

Exogenous capture of visual spatial attention by olfactory-trigeminal stimuli 

Dear Dr. Delplanque:

I'm pleased to inform you that your manuscript has been deemed suitable for publication in PLOS ONE. Congratulations! Your manuscript is now with our production department. 

Kind regards, 

on behalf of

Dr. Jessica Freiherr 

Academic Editor

PLOS ONE